# Determining Exception Context in Assembly Operations from Multimodal Data

**DOI:** 10.3390/s22207962

**Published:** 2022-10-19

**Authors:** Mihael Simonič, Matevž Majcen Hrovat, Sašo Džeroski, Aleš Ude, Bojan Nemec

**Affiliations:** 1Department of Automatics, Biocybernetics and Robotics, Jožef Stefan Institute, Jamova Cesta 39, 1000 Ljubljana, Slovenia; 2Faculty of Electrical Engineering, University of Ljubljana, Tržaška Cesta 25, 1000 Ljubljana, Slovenia; 3Department of Knowledge Technologies, Jožef Stefan Institute, Jamova 39, 1000 Ljubljana, Slovenia; 4Jožef Stefan International Postgraduate School, Jamova Cesta 39, 1000 Ljubljana, Slovenia

**Keywords:** sensor fusion, predictive clustering trees, autonomous exception handling, autonomous assembly, peg-in-hole

## Abstract

Robot assembly tasks can fail due to unpredictable errors and can only continue with the manual intervention of a human operator. Recently, we proposed an exception strategy learning framework based on statistical learning and context determination, which can successfully resolve such situations. This paper deals with context determination from multimodal data, which is the key component of our framework. We propose a novel approach to generate unified low-dimensional context descriptions based on image and force-torque data. For this purpose, we combine a state-of-the-art neural network model for image segmentation and contact point estimation using force-torque measurements. An ensemble of decision trees is used to combine features from the two modalities. To validate the proposed approach, we have collected datasets of deliberately induced insertion failures both for the classic peg-in-hole insertion task and for an industrially relevant task of car starter assembly. We demonstrate that the proposed approach generates reliable low-dimensional descriptors, suitable as queries necessary in statistical learning.

## 1. Introduction

Assembly tasks, such as inserting parts into fixtures, are among the most common industrial applications [1]. Robot assembly typically requires a good understanding of the procedure and knowledge about part properties and geometry [2]. Therefore, most of the deployed robotic systems used today are carefully programmed [3]. As such, they are limited to performing a specific assembly task in structured environments without external disturbances. Nevertheless, they can fail due to various errors that cannot be foreseen in advance. Possible causes include deviations in the geometry of the workpiece, imprecise grasping, etc. In such cases, it is necessary for the operator to manually eliminate the cause of the error, reset the system, and restart the task [4]. Current robotic systems do not learn from such situations. If a similar situation repeats, human intervention is needed again. To ensure robust execution of robot assembly tasks, it is increasingly important to handle such exception scenarios autonomously, possibly by incorporating previous experience.

The first step toward building such an autonomous system is to determine the reason for the failure. For example, a robot may fail to assemble two parts, but it is unclear whether it has failed because the parts do not match or because of an ineffective manipulation strategy [5]. Understanding or at least classifying the reason for the failure is, therefore, crucial for the successful design of a corresponding exception policy.

Recently, we have proposed a framework for the learning of exception strategies [6], which is based on determining the context of the failure. The extracted context is associated with different robot policies needed to resolve the cause of the error. In the event of an error, the system stops, and the robot switches to gravity compensation mode. Using incremental kinesthetic guidance [7], the operator performs a sequence of movements to allow the continuation of regular operation. First, the robot builds a database of corrective actions and associates them with the detected error contexts. Then, using statistical methods, it computes an appropriate action by generalizing the corrective actions associated with different error contexts. In this way, the robot becomes increasingly able to resolve errors on its own and eventually does not require human intervention to resolve assembly failures anymore (see Figure 1).

Modern robotic systems are equipped with a wide variety of sensors that can be used to detect a possible failure. On the other hand, context determination can be seen as inferring the circumstances that have resulted in the given outcome. The first step in the exception strategy learning framework is, therefore, context determination, which can be seen as inferring a minimal representation of the circumstances that have resulted in the given outcome. As raw sensor data are usually high-dimensional, they cannot be directly used with statistical learning methods that allow us to relate the observed state (context) to the previous states and generate an appropriate robot action to resume regular operation. Moreover, for reliable context determination, it is often necessary to combine complementary information from different sensor modalities. This process is known as data fusion and can lead to improved accuracy of the model compared to a model based on any of the individual data sources alone [8]. Ensemble learning methods have proven to be appropriate for addressing multimodal classification and regression problems in many domains [9].

We propose to train models that generate low-dimensional context descriptions based on multimodal sensor data from vision systems and force-torque sensors. In this sense, context determination can be defined as determining of the type of circumstances from multimodal data. We use an intermediate-fusion approach, where we first extract modality-specific features, as shown in Figure 2. We rely on a state-of-the-art neural network model for image segmentation to extract features from images, whereas we use contact point estimation to extract data from the measured forces and torques. To generate a low-dimensional context description of the circumstances that resulted in the given outcome from the extracted features, we use ensembles of predictive clustering trees (PCTs) [10], which are well suited for handling hierarchical multi-label classification (HMLC) tasks. With the proposed hierarchical approach, the training of the context estimation model can be divided into multiple phases, allowing for an incremental approach. The time-consuming training of the image segmentation model only needs to be performed once, whereas fast training of the high-level ensemble model can be performed each time a new class needs to be added.

The evaluation of the proposed approach comprised two scenarios. Peg-in-hole assembly is chosen as the first use case because it reflects the typical complexity of industrial assembly tasks [13]. We show that it is possible to apply the method to other tasks by performing an evaluation of a car starter assembly task. The approach can also be applied to further situations. Apart from identifying error classes, the only process-specific step is selecting the image segments of interest and training the instance segmentation model accordingly.

The main highlights of our context determination approach are:with the use of multimodal data we get an improved predictive performance of ensembles;it is easy to add new classes (this is necessary as we discover new failure cases incrementally as they arise);the approach generalizes well to new cases (we can make useful predictions based on a model trained on a limited amount of data).

The paper is organized into six sections. Section 2 reviews current strategies to handle exceptions in robot assembly and the usage of tactile and vision sensor data in robotics. The details of our approach are presented in Section 3. In Section 4, the predictive performances of different variations of the developed model for context determination are evaluated. Section 5 discusses the results as well as future plans. We conclude with a brief summary of the paper in Section 6.

## 2. Related Work

Fault-tolerant robotic systems that are able to detect and autonomously deal with system failures have been the subject of research for many years [14]. While some researchers are concerned with fault tolerance in medical, space, nuclear, and other hazardous applications, our research focuses on industrial processes, where we can ensure operator presence, at least in the learning phase. In such environments, strategies based on various heuristic movement patterns (random search, spiral search, dithering, vibrating, etc.) are often used to deal with unexpected situations [15,16].

Laursen et al. [17] proposed a system that can automatically recover from certain types of errors by performing the task in reverse order until the system returns to a state from which the execution can resume. Error recovery can also be performed collaboratively so that the robot recognizes when it is unable to proceed and asks for human intervention to complete the task [18]. Recently, it was proposed to use ergodic exploration to increase the insertion task success rate based on information gathered from human demonstrations [19]. Another method exploits variability in human demonstrations to consider task uncertainties and does not rely on external sensors [2].

On the other hand, another line of research highlights the importance of sensorimotor interaction for future learning methods in robotics [20]. During the assembly task execution, monitoring the exerted forces and torques is necessary to prevent damaging the parts or robot [6]. These data can be used to avoid large impact forces exploiting compliance and on-line adaptation [21], to speed up the process in the subsequent repetitions [22], to determine contact points and learn contact policies [11], and to predict [23] or classify [24] robot execution failures.

Various previous works have studied the idea to calculate contact points based on force-torque measurements [11,25,26,27]. In our previous work [6], we used force-torque measurements to calculate trajectory refinements that enable successful insertion despite the grasping error. Force-torque data carry enough information to generate an appropriate refinement, given that we already know the nature of the problem (orientation vs. position grasping error). However, in general, sufficient information cannot be obtained from force-torque data only. For example, the policies for correcting unsuccessful assembly attempts often depend on which part of the peg is in contact with the environment [28]. Thus additional sensors are required. Using a force-torque sensor only is also problematic due to sensor noise. Many applications in process automation, therefore, rely on vision systems to extract the necessary information. While vision can be quite sensitive to calibration errors and typically requires a well-designed workcell to ensure optimal lighting conditions and avoid occlusions, it is fast and can be used for the global detection of multiple features [29].

The advantage of combining visual and contact information has been investigated in multiple works in robotics over a longer period of time. This research includes dimension inspection [30], object recognition [31], and localization [29]. In the context of robot assembly, visual and tactile sensing has been used to continuously track assembly parts using multimodal fusion based on particle filters [32] and Bayesian state estimation [13]. We share the principal idea of combining data from visual and force-based sensing. We want to further develop these concepts towards structured representations of the task context in order to develop an integrated solution for the automatic handling of failures in assembly processes.

Multimodal fusion combines information from a set of different types of sensors. Detection and classification problems can be addressed more efficiently by exploiting complementary information from different sensors [8]. Different methods for data fusion from multimodal sources exist. Generally, we can distinguish three levels of data fusion: early fusion, where the raw data are combined ahead of feature extraction and the result is obtained directly; intermediate fusion, where modality-specific features are extracted and joined before obtaining the result; and late fusion, where the modality-wise results are combined [33,34]. It may seem that combining multimodal data at the raw data level should yield the best results, as there would be no loss of information. However, due to the unknown inter-dependencies in raw data, fusion at a higher level of abstraction may be a more helpful approach in practice [35].

Ensemble learning is a general approach in machine learning that seeks better predictive performance by combining the predictions from multiple models [9]. Ensemble learning methods have proven to be an appropriate tool to address multimodal fusion, achieving comparable results or even outperforming other state-of-the-art methods in many other domains [36,37]. The idea of ensemble learning is to employ multiple models and combine their predictions. This is often more accurate than having a complex individual model to decide about a given problem. Data from heterogeneous sources, such as different modalities [38], can easily be combined. In this paper, we consider an ensemble of predictive clustering trees (PCTs) [10] to perform hierarchical multi-label classification (HMLC). PCTs are a generalization of ordinary decision trees and have been successfully used for a number of modeling tasks in different domains, i.e., to predict several types of structured outputs, including nominal/real value tuples, class hierarchies, and short time series [39,40]. A detailed description of PCTs for HMLC is given by Vens et al. [41].

## 3. Materials and Methods

In this section, we first describe the robotic workcell used to collect the data described in Section 3.1. The assembly tasks to perform the evaluation are presented in Section 3.2 and Section 3.3. Next, we present our contact determination method in detail. The approach consists of three main parts. First, force-torque sensor measurements are processed using a method for contact point estimation described in Section 3.4. Data from the vision modality are passed through a neural-network model performing instance segmentation (Section 3.5), and features are extracted from the instance masks using standard computer vision methods (Section 3.6). Finally, the features from both modalities are combined using an ensemble of predictive clustering trees, as described in Section 3.7.

### 3.1. Experimental Environment

In our research, we focused on robotic assembly and considered two tasks—square peg insertion using the Cranfield benchmark [42] and the industrially relevant car starter assembly [43]. To perform both tasks and collect data for the context determination model, we rely on a modular workcell design that enables easy mounting of task-specific equipment, e.g., robots, sensors, and auxiliary devices [44] and a ROS-based software architecture that allows for easy integration of new components [45].

The workcell consists of two modules and a control workstation. The first module supports a seven-degree-of-freedom collaborative robot, Franka Emika Panda. The other module is equipped with sensors and cameras to support the specific assembly process, as shown in Figure 3. An Intel RealSense D435i RGB-D camera is used to supervise the insertion visually. To control the light conditions, we utilize an adaptive lighting setup based on Aputure Amaran F1 LED panels. Besides images, we can also capture forces and torques. To measure the forces exerted in the peg-in-hole task, we utilize an ATI Delta force-torque sensor mounted under the Cranfield benchmark plate. To measure the forces exerted in the copper ring insertion task, we utilize a wrist-mounted ATI Nano25 sensor. Additional peripheral devices, visible in Figure 3b, are used to assist different aspects of the human–robot collaboration, which is not the subject of this paper.

To perform the assigned tasks, we applied a passivity-based impedance controller for manipulators with flexible joints [46]. We assume that the controller parameters were carefully tuned to ensure stable and compliant operation in unstructured environments, where we can expect deviations in task parameters.

### 3.2. Peg-in-Hole Insertion Task

The peg-in-hole (PiH) task is an abstraction of the most typical task in assembly processes, accounting for approximately 40% of the total assembly tasks [47]. Over time, many different approaches and control strategies to address this problem have emerged. Nowadays, the efficiency of the applications is enhanced by integrating machine vision and other sensor technology accompanied by artificial intelligence approaches. As such, it is a commonly accepted benchmark in assembly research.

To generate a dataset for comparing different methods for failure context determination, we repeatedly executed the task of square peg insertion using the Cranfield benchmark. It requires the insertion of a square peg into the corresponding hole of the base plate. The main challenge is the transition of the peg from free space into a highly constrained target hole. Relatively tight tolerances combined with imprecise positioning can prevent the successful completion of the insertion process.

Different factors influence the outcome of the PiH task. For instance, both imprecise grasping and wrong target position can lead to insertion failure, as shown in Figure 4.

In order to collect a database of different insertion outcomes, we deliberately set different positional offsets in either the *x* or the *y*-direction from −10 to 10 mm in 1 mm steps. In this way, we generated 40 cases that resulted in insertion failure and 1 that led to successful insertion. Due to the offset, the robot fails to insert the peg and stops the execution when it exceeds a force threshold, set to 10 N in the *z*-direction. The insertion is successful when there is no positional offset in either direction.

In total 180 data entries were recorded. The robot attempted to insert the peg into the plate three times for each failure case. Additionally, 60 successful attempts were recorded. Robot pose, force and torque measurements, and RGB images of the outcome were captured when the insertion was complete or stopped (force threshold exceeded).

Note that the data can be organized hierarchically into three categories (no error, positional error in *x* direction, and positional error in *y* direction). The latter two categories can be further split in half depending on the direction of the error (*x*/+, *x*/−, *y*/+, *y*/−). Finally, we can split based on the magnitude of the error (e.g., *x*/+/2, meaning that we have a 2 mm error in the x+ direction).

### 3.3. The Task of Inserting Copper Sliding Rings into Metal Pallets

The car starter assembly process includes inserting copper sliding rings into metal pallets, as shown in Figure 5. This can be categorized as a multiple peg-in-hole problem, as it is necessary to insert the bottom of the copper ring and both upper part lugs correctly. The insertion process is challenging due to the deformability of the sliding rings. This task has been taken from a real production process where it is performed manually. Previous automation attempts have failed due to the low success rate that was achieved. To ensure robust insertion, we proposed to use the exception strategy framework [43].

We have collected a database of twelve copper ring insertions, which includes both successful insertions and deliberately induced insertion failures. The failures were caused by the displacement of the target position for insertion in the *x* and *y* directions:no displacement, leading to successful insertion;positional displacement in the *x* direction, with $\Delta p_{x}$ between 1 and 3 mm in 1 mm steps;positional displacement in the *y* direction with $\Delta p_{y}$ between 1 and 3 mm in 1 mm steps, both leading to unsuccessful insertion.

Additionally, we recorded insertion attempts with deformed parts, which also led to an unsuccessful insertion. For each of the cases, we recorded at least four insertion attempts. The process was repeated for all four slot positions in the molding cast. Each entry consists of a snapshot of the outcome of the insertion task (cropped RGB image) and the time series of force $F=(F_{x},F_{y},F_{z})$ and torque $T=(T_{x},T_{y},T_{z})$ measurements. Similarly to the PiH dataset, the gathered data can be organized hierarchically.

### 3.4. Force-Torque Data Extraction: Contact Vector Estimation

In our previous work [6], we have considered only errors due to the offset in the grasping angle and have shown that force-torque data can be used to determine a suitable context descriptor using principal component analysis (PCA), which correlated most strongly with the grasping error. Such a dimensionality reduction is beneficial because the generation of an appropriate refinement trajectory based on statistical learning is sensitive to the dimension of the feature space. Another possible approach to reduce the dimensionality of force-torque measurements is to determine contact points [48].

The point of contact between two parts can be estimated based on the relationship between force F, torque T, and lever r by using the following formulation [11]
(1)$$r\left(\alpha \right)=\frac{F\times T}{{∥F∥}^{2}}+\alpha \frac{F}{∥F∥},$$
where α is a suitably chosen constant so that the vector r touches the environment as illustrated in Figure 6a. The measured forces and torques must be expressed in the robot end-effector coordinate system.

However, the contact point estimation cannot always distinguish between the different types of errors, as illustrated in Figure 6b. Thus, forces and torques, as well as the positional data, cannot uniquely determine the context. In order to resolve this ambiguity, we introduce another modality, as discussed in the remainder of this paper.

Nevertheless, Equation (Equation 1) provides a suitable representation that can distinguish between different conditions of the same outcome type. A graphical example of contact vector estimation for both experiments is shown in Figure 7.

Our preliminary results have shown that the inclusion of raw force-torque data as features decreases the performance of the final model. Thus the feature vector for the FT modality was chosen to include only the contact point vector estimate. For each example k∈E, the feature vector is calculated as:(2)$$f_{\mathrm{FT}}^{k}=\left(r_{x},r_{y},r_{z}\right),$$
where $r_{x},r_{y},r_{z}$ are components of the vector r(α).

### 3.5. Vision Data Extraction: Instance Segmentation with YOLACT

We applied deep neural networks (DNN) to perform feature extraction from image data. They provide good flexibility because pre-trained NN models and frameworks can be re-trained by using a custom dataset for a specific use case, in contrast to the classic computer vision (CV) algorithms, which tend to be more domain specific [49]. Compared to the traditional computer vision methods (e.g., edge detection), they often require less manual fine-tuning.

Various convolutional neural networks (CNNs) have proven to be suitable for analyzing image data. An essential issue with a custom network that directly extracts features is that retraining is needed when a new error class is added or the camera position is changed. For these reasons, we rely on models that are designed to be less prone to changes in object position in the picture. This has been extensively studied in object detection and instance segmentation models. Instance segmentation is an enhanced type of object detection that generates a segmentation map for each detected instance of an object in addition to the bounding boxes.

In order to meet the above-listed requirements, we used the state-of-the-art instance segmentation model YOLACT [12]. YOLACT builds upon the basic principles of RetinaNet [50] with the Feature Pyramid Network [51] and ResNet-101 [52] as a convolutional backbone architecture for feature extraction. It utilizes a fully convolutional network to directly predict a set of prototype masks on the entire image. Lastly, a fully connected layer assembles the final masks as linear combinations of the prototype masks, followed by bounding box cropping. Compared to most of the previous instance segmentation approaches, such as Mask R-CNN [53], which are inherently sequential (the first image is scanned for regions with object candidates, then each of them is processed separately), YOLACT is a one-stage algorithm that skips this intermediate localization step. This allows for nearly real-time performance. By using shallower computational backbones, such as ResNet-50, even faster performance can be achieved at a minimal accuracy cost when compared to ResNet-101 [12].

The (re)training of YOLACT requires labeled images and ground truth image masks. We have used an open-source graphical image annotation tool, Labelme, to annotate images in our datasets (https://github.com/wkentaro/labelme, accessed on 13 October 2022). For the PiH dataset, we manually annotated 30 images for each position using four different light settings. We manually annotated 10 images for each position using two different light settings for the copper ring insertion dataset. We split the annotated datasets into training and validation partitions, with 80% and 20% of the data, respectively. Finally, the annotations had to be transformed into a format compatible with the YOLACT training script (COCO). For this purpose, we prepared an open-source tool—labelme2coco (https://github.com/smihael/labelme2cocosplit, accessed on 13 October 2022).

In the proposed pipeline, we configured YOLACT to use a computationally lighter ResNet-50 as the backbone. This enabled us to use original-resolution images while retaining high training and inference speed. We trained two models for each of the above-presented datasets. During training, the algorithm used a batch size of 8, weight decay of 0.0005, and image size of 1280 × 720 pixels (PiH dataset) or 221 × 381 pixels (copper ring insertion dataset). The initial learning rate was set to 0.001. The model was trained for 40,000 iterations, and the decay rate of 0.1 was applied once each 10,000 iterations. The training took 8 h on a GeForce GTX 1060 GPU for the PiH dataset and 6 h for the copper ring insertion dataset.

Once the models were trained, we deployed them to a workstation in the robotic workcell. The integration was done using a modified yolact_ros package (https://github.com/smihael/yolact_ros, accessed on 13 October 2022), which also allows using the learned model for inference without a GPU, thus lowering the computational requirements.

The results are shown in Figure 8. The PiH model is trained to distinguish between the peg and the base plate, whereas the copper ring insertion model is able to distinguish the following segments: gripper, mold, screws, ring base, wings, and ears (lugs).

Note that the PiH model can be equally used for any of the two insertion slots in the PiH task. Likewise, the copper rings model can be used for any of the four slots in the copper ring insertion task. Since the model is position invariant, meaning that the model is able to correctly mark the area of different image parts regardless of where in the image they appear, we can apply it for the analysis of new error cases.

### 3.6. Extracting a Fixed-Size Feature Vector from Instance Segmentation Results

Using the trained YOLACT segmentation models, we obtain bounding boxes and image masks for all images in the PiH and copper ring insertion datasets. The information obtained from instance segmentation needs to be further processed to be used in further steps of the pipeline, as shown in Figure 9. We extract fixed-size feature vectors, as ensembles of predictive clustering trees do not operate over image masks.

An image is represented as a w×h×3 matrix of pixels I(x,y,c)∈{0,1,…,255}, representing the RGB color channels. The image can contain multiple instances of different objects. For each segmented object instance *s*, we obtain its type, the bounding box, and the mask. The bounding box $B_{s}$ is given as a pair of pixel coordinates of two diagonal corners $\left\{\left(x_{1},y_{1}\right),\left(x_{2},y_{2}\right)\right\}$. The bounding box can be represented by the centroid $c_{s}$, width $w_{s}$, and height $h_{s}$ of the rectangle
(3)$$\begin{array}{ccc} c_{s} & = & {\left[c_{s,x},c_{s,y}\right]}^{⊤}={\left[\frac{x_{1}+x_{2}}{2},\frac{y_{1}+y_{2}}{2}\right]}^{⊤}, \end{array}$$
(4)$$\begin{array}{ccc} w_{s} & = & x_{2}-x_{1}, \end{array}$$
(5)$$\begin{array}{ccc} h_{s} & = & y_{2}-y_{1}. \end{array}$$

The pixels belonging to the specific object instance *s* are represented with masks. Each mask is a w×h binary matrix $M_{s}\in B^{w\times h}$, which tells whether a pixel is part of the mask or not. Using PCA, we determine the first principal component for each instance’s mask $e_{s,1}=\left(x,y\right)$. This result can be used to calculate the orientation of the part in the image plane (visualized in Figure 10)
(6)$$\varphi_{s}=\arctan\left(\frac{-y}{x}\right)-\frac{\pi }{2}.$$

Additionally, we calculate the pixel area of each instance mask as a total number of all true elements in the instance matrix
(7)$$a_{s}=\sum_{x=0}^{w}\sum_{y=0}^{h}m_{s}\left(x,y\right).$$

In both experiments, we define a set of object of interests $S_{\mathrm{int}}$ (see Figure 8). For the peg-in-hole insertion task, it consists of the peg only, while for the copper ring insertion task, $S_{\mathrm{inst}}$ contains the gripper, ring, wings, and ear. Note that additional features, e.g., screws on the molding cast or the base of the Cranfield benchmark, can be used as calibration features. In our case, this was not needed as the datasets were recorded with a fixed camera position.

From the set of examples with no visible errors $E_{0}$, we calculate the average segment mask ${\bar{M}}_{s}$ for each object instance *s* of interest from $S_{\mathrm{int}}$. The average mask is calculated as the element-wise mean of the mask matrices:(8)$${\bar{M}}_{s}=\frac{1}{|E_{0}|}\sum_{k\in E_{0}}M_{s}^{k}\in R_{+}^{w\times h}.$$

For other examples, we compute the difference between theirs masks and the average mask of examples with no error ${\tilde{M}}_{s,\mathrm{diff}}^{k}=M_{s}^{k}-{\bar{M}}_{s}$, and take only its positive elements to define binary matrix $M_{s,\mathrm{diff}}^{k}$,
(9)$$m_{s,\mathrm{diff}}^{k}\left(x,y\right)=\left\{\begin{array}{c} 1,\;{\tilde{m}}_{s,\mathrm{diff}}^{k}\left(x,y\right)>0\\ 0,\;\mathrm{otherwise}\; \end{array}\right..$$

An example is shown in Figure 11. For each of the obtained difference segments, we then calculate its center $c_{s,\mathrm{diff}}={\left[c_{s,\mathrm{diff},x},c_{s,\mathrm{diff},y}\right]}^{⊤}$ and pixel area $A_{s,\mathrm{diff}}$ using Equations (Equation 3) and (Equation 7), respectively.

In this way, we obtain an image feature vector for each example k∈E and object of interest $s\in S_{\mathrm{int}}$:(10)$$f_{s}^{k}={\left[c_{s}^{k\;⊤},\;w_{s}^{k},\;h_{s}^{k},\;\phi_{s}^{k},\;a_{s}^{k},\;c_{s,\mathrm{diff}}^{k\;⊤},\;a_{s,\mathrm{diff}}^{k}\right]}^{⊤}.$$

### 3.7. Combining Image Features and Force-Torque Measurements Using Ensembles of Predictive Clustering Trees

We formulate the determination of the outcome of the insertion task as a hierarchical multi-label classification (HMLC) problem. Given the extracted image features and the estimate of the contact point, the type of outcome should be predicted. For the different types of outcomes, a hierarchy of class labels defines the direction and magnitude of the underlying error, as described below.

We applied ensembles of predictive clustering trees (PCTs) [10] for this task. PCTs are a generalization of ordinary decision trees [41]. Generally, in a decision tree, an input is entered at the top and as it traverses down the tree, the data gets bucketed into smaller and smaller sets until the final prediction can be determined. The PCT framework, however, views the decision tree as a hierarchy of clusters: the top node corresponds to the cluster containing all of the data, which is recursively partitioned into smaller clusters so that per-cluster variance is minimized [39]. In this way, cluster homogeneity is maximized, and consequently, the predictive performance of the tree is improved.

PCT ensembles consist of multiple trees. In an ensemble, the predictions of classifiers are combined to get the final prediction. For an ensemble to have better predictive performance than its individual members, the base predictive models must be accurate and diverse [9]. The diversity between trees in the PCT framework is obtained by using multiple replicas of the training set and by changing the feature set during learning, as in the random forest method [54].

In our setting, each example *k* from the set of examples E consists of all extracted features $f_{k}$ and the corresponding label vector $l_{k}$. The feature vectors are obtained by concatenating per-modality features:(11)$$f_{k}={\left[{f_{\mathrm{FT}}^{k}}^{⊤},\;{f_{1}^{k}}^{⊤},\;{f_{2}^{k}}^{⊤},\;\ldots ,\;{f_{\left|S\right|}^{k}}^{⊤}\right]}^{⊤},$$
with $f_{\mathrm{FT}}^{k}$ and $f_{s}^{k},\;s=1,\ldots ,\left|S\right|,$ defined as in Section 3.4 and Section 3.6, respectively. To define the corresponding label vector, we first observe that in HLMC, each example can have multiple labels. Classes are organized in a hierarchical structure, i.e., an example belonging to a class also belongs to all of its superclasses. The resulting ordered set of classes is used to define a binary label vector $l_{k}\in B^{L}$. The components of $l_{k}$ are equal to 1 if the example is labeled with the corresponding class and 0 otherwise. *L* denotes the number of all classes in the hierarchy.

For the PiH task, the set of labels at the first hierarchical level consists of “no error”, and “x” and “y” for the error in one of the two directions. At the second level, we have “x+” and “x−” as subclasses of “x”, and “y+” and “y−” as subclasses of “y”. Likewise, we have “x + 1”, “x − 1”, “y + 1”, “y − 1”, “x + 2”, …, “y − 10” at the third hierarchical level. For the copper ring insertion task, the set of labels at the first hierarchical level consists of: “no error”, “bad part”, and “x” and “y” for the error in one of the two directions. Similarly, as in the PiH task, the sub-classes at the second and third levels are representing various magnitudes of error (ranging from −10 to 10 mm in 1 mm steps) in both considered directions (*x* and *y*).

In summary, to train the ensemble of predictive trees, we collect the dataset E
(12)$$E={\left\{f_{k},\;l_{k}\right\}}_{k=1}^{K}.$$

After training we can use the resulting ensemble of predictive trees to predict the labels l given the extracted feature vector f.

We trained multiple ensembles for two different tasks. See Section 4 for more details. We used PCT ensembles, i.e., random forests of PCTs, as implemented in the CLUS system (CLUS is available for download at http://source.ijs.si/ktclus/clus-public, accessed on 13 October 2022) for this purpose. Each ensemble consisted of 50 trees. As a heuristic to evaluate the splits in decision trees, we used the variance reduction [39]. The variance for the set of examples E is defined as the average squared distance between each example’s label vector $l_{k}$ and the set’s mean label vector $\hat{l}$, i.e.,
(13)$$\mathrm{Var}\left(E\right)=\frac{1}{\left|E\right|}\sum_{k\in E}d{\left(l_{k},\hat{l}\right)}^{2}$$

The distance measure used in the above formula is the weighted Euclidean distance:(14)$$d\left(l_{1},l_{2}\right)=\sqrt{\sum_{i=1}^{L}w\left(c_{i}\right){\left(l_{1,i}-l_{2,i}\right)}^{2}},$$
where the class’s weight $w(c_{i})$ depends on its depth within the hierarchy. The similarities at higher levels in the hierarchy are considered more important than the similarities at lower levels. Therefore, the class weights $w(c_{i})$ decrease with the depth of the class in the hierarchy. $w(c_{i})$ is typically set as $w_{0}^{d}$, where *d* is the depth of the label in the hierarchy: $w_{0}$ was set to 0.75 in our experiments. The number of randomly selected features at each node was set to $\lfloor \sqrt{L}\rfloor +1$, where *L* is the total number of features.

To combine the predictions of all classifiers in the ensemble and obtain the final prediction, their average is taken.

## 4. Results

In this section, we evaluate the performance of our proposed approach along two dimensions: its generalizability to handle unseen data and the effect of including/excluding features from separate modalities. Finally, the setup was experimentally verified in the robotic workcell.

### 4.1. Generalizability of Classification

In order to verify how well the approach can generalize to unseen data, we train a model on a subset where we do not include any examples of a particular outcome case. Since the database for the copper ring insertion task is not sufficiently fine-grained, this aspect was evaluated only for the peg-in-hole task. We excluded all cases where the positional error in any direction equals 5 mm and observed if the model could correctly predict the direction of error for the excluded examples. The results are shown in Figure 12. The model correctly predicted the direction of error for all the excluded examples, both at the first and the second level of hierarchy. As the predictions at the third hierarchical level describe the magnitude of error, they can also be evaluated using root mean square error (RMSE). At the third hierarchical level, it assigned all the excluded examples to the closest lower error class that was presented in training for the *x* direction (RMSE${}_{x}=1$ mm), whereas for the *y* direction it did so for 4 of the 6 examples (resulting in RMSE${}_{y}=1.91$ mm).

### 4.2. Single Modality versus Multimodal Models for Classification

We assessed the effectiveness of including/excluding features from the individual modalities by training multiple PCTs on different subsets of features for both tasks:only features based on the image data (see Section 3.6);only features based on the force-torque sensor data (see Section 3.4);features from both modalities.

Models were trained using 80% of the data and tested on the remaining 20% of the data.

The results for the PiH task are given in Figure 13. We found out that the model performed best when all features were included, indicating that the features from both modalities are complementary and improve the model’s performance. The overall classification accuracy was calculated for the multi-class classification problem by taking the sum of the true positives and true negatives for each label, divided by the total number of predictions made. The accuracy was then averaged by support (the number of true instances for each label). For the model that uses all features, the overall classification accuracy at the first hierarchical level was 0.98. At the second level, the accuracy was 1.0. For the third level, the overall classification accuracy was 0.68. For error classes in the *x* and *y* directions, the classification accuracy was 0.55 and 0.5, respectively. For the model that only uses features from the vision modality, the overall classification accuracy at the first two hierarchical levels stayed the same, indicating that vision features can distinguish well between different types of outcomes. The overall classification accuracy at the third level was 0.68, and 0.5 and 0.55 for the *x* and *y* directions, respectively. When evaluating the model that only uses features from the FT modality, the overall classification accuracy at the first level dropped to 0.92, at the second to 0.95, and at the third to 0.61, whereas it was 0.5 and 0.35 for the *x* and *y* directions, respectively.

The results for the copper ring insertion task are given in Figure 14. Similar to before, we found that the model performed best when all features were included. For the model that uses all features, the overall classification accuracy at the first two hierarchical levels was 0.88. For the third level, the overall classification accuracy was 0.81. For error classes in *x* direction, the classification accuracy is 0.67, and 0.9 for *y* direction. For the model that only uses features from the vision modality, the overall classification accuracy at the first two hierarchical levels dropped slightly to 0.85. The overall classification accuracy at the third level was 0.62, and 0.5 and 0.6 for the *x* and *y* directions, respectively. The drop was even more pronounced when evaluating the model that only uses features from the FT modality. The overall classification accuracy at the first level was 0.81 and at the second and third it was 0.77, whereas it was 0.58 and 0.9 for the *x* and *y* directions, respectively. When comparing the results of the vision- and FT-features-only models, it is evident that while the earlier achieved a higher overall accuracy, the latter achieved higher accuracy when distinguishing among different magnitudes of error in the *y* direction.

### 4.3. Verification of Error Context Determination for the Generation of Exception Strategies

The proposed framework was experimentally verified on both the PiH and the copper ring insertion task. The initial PiH policy was carefully programmed and executed in the workcell with the same setup as described in Section 3.2. In order to cause an exception, the target position was displaced by 6 mm. The proposed approach correctly estimated the error context to “x/-/6”. Since the exception strategy for this case has not yet been programmed, the robot stopped and prompted the operator. Using kinesthetic guidance, the operator guided the robot back along the policy to an appropriate point, where it is possible to resume the operation. The operator then demonstrated the correction, which resolved the problem. When we displaced the target position by 6 mm again, which resulted in a similar outcome, the robot again classified it as “x/-/6”. As the exception strategy is now known, the robot could resolve the situation using the demonstrated exception strategy. In a similar manner, the operator demonstrated policies for the case where the target was displaced by 4 mm in a positive *x* direction. When we displaced the target position by 5 mm, the robot correctly classified the context to be “x/-”, whereas the magnitude was not determined precisely (4 mm) as we used the model that did not include error contexts of this magnitude in the training set. Nevertheless, by combining the policies demonstrated for the other two cases in the “x/-” category and using locally weighted regression, as proposed in [6,43], the robot was able to perform the insertion successfully.

When inserting a sliding ring into the casting mold, there are two major types of errors. The first type is when the base of the ring is not properly seated into the model (see Figure 5 middle). This type of error mainly arises due to imprecise grasping or due to errors in the target position. The second type of error occurs when the sliding ring is deformed. Both types of error can be reliably determined by using the proposed approach. We first displaced the target position by 2 mm in the negative *y* direction so that the insertion failed. As the exception strategy for this case has not yet been programmed, the operator demonstrated how to resume the operation and resolve the issue using iterative kinesthetic guidance [7]. When the target was displaced by the same offset again, the robot was able to resolve the problem. We also started the insertion procedure with a deformed part. It was correctly determined, and the robot placed it into the bin for deformed parts. An example video of both experiments can be found in the Appendix A.

Note that the application of the exception strategy learning framework does not affect the cycle time in successful attempts. Once the model is deployed, the time to obtain context estimation is negligible. In unsuccessful attempts, where an alternate policy needs to be demonstrated or executed, the cycle time is, however, prolonged. However, since these situations are less frequent, this has very little effect on the average cycle time of an automated line.

## 5. Discussion

The results of our study indicate that the application of multimodal features leads to an improved classification accuracy of the ensemble models employed for classification. This implies that the features are complementary and taken together provide greater discrimination power than the features stemming from a single modality. Prediction errors that arose when applying vision-only-based models, showed the limitations of two-dimensional image data, thus depth information should be considered in the future.

It is important to note that, to a large extent, the models were able to correctly assign error types to examples with a magnitude of error not included in the training data. This is a critical finding as it indicates that the computed models are robust and can be used in real-world applications, also in less-structured non-industrial environments, where error types can not be predicted in advance. To evaluate this aspect, we excluded all examples with a certain magnitude of error from the training set. The results show that the models still perform well, indicating that they are not overfitting the training data.

Based on the observation that features obtained from different modalities contributed to improving the classification results at different levels, a more explicit hierarchical pipeline could be considered in the future, exploiting the robot as an agent that can interact with the environment. Data from different modalities would contribute towards the final prediction at different stages of the process, consisting of, e.g.,

(1)the type of error (due to positional displacement, part geometry, imprecise grasping), determined based on image data;(2)the magnitude of error, based on force-torque or depth data.

The context determination does not have to occur instantaneously but can include exploring the environment as part of the pipeline. We could first use the vision data to determine the direction in which the robot should move in order to reduce the error (left/right). The robot can then move in this direction until it detects a new state (one of the force-torque components changes or a compliant robot stops moving as it hits an obstacle—see [55]). In the newly found state, the robot again estimates the direction in which it needs to continue or stop.

In the future, we intend to expand the proposed approach by considering other possible error types (e.g., arising from erroneous orientation when grasping) and their combinations (displacements in multiple degrees of freedom at the same time), as well as properly handling continuous data (regression at the lower hierarchical level instead of classification). We believe that the presented framework is not only applicable to learning error context but could also be extended to cognitive systems that will be able to respond autonomously to changes in the environment. To achieve these goals, the improved versions of our approach should consider additional modalities and alternative features extraction methods.

## 6. Conclusions

In this work, we have proposed a novel method for context determination based on multimodal features that can be used for learning exception strategies in various assembly tasks.

Our approach was validated on two tasks, the classic peg-in-hole, and the copper ring insertion. To evaluate its effectiveness, we deliberately induced different types of errors, which led to failed task executions. Using the proposed approach, the error type was correctly obtained in all cases, allowing for correction of the task execution parameters and finally leading to successful task performance.

The study results indicate that the features used in the ensemble models are complementary and that the multimodal setup achieves the highest classification accuracy. Moreover, the model can correctly assign error types to examples with an unknown error magnitude.

In the current implementation, the context was calculated based on the measurement of forces and torques and RGB sensor data. The introduction of further sensor modalities (such as depth data) could lead to a further increase in classification accuracy.

## Figures and Tables

**Figure 1 sensors-22-07962-f001:**
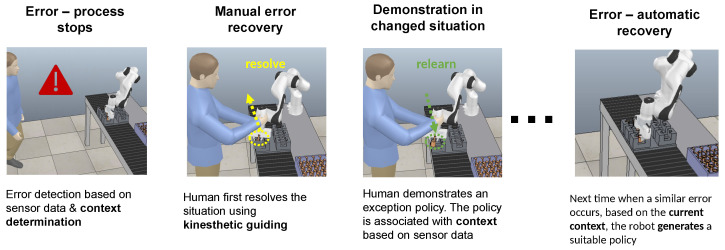
Simplified exception handling workflow [6]. This paper is about context determination, which is an essential requirement of the workflow.

**Figure 2 sensors-22-07962-f002:**
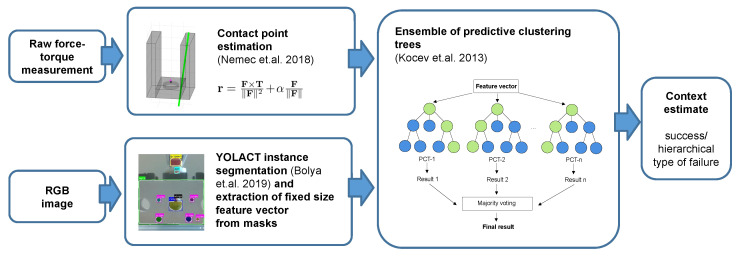
Context determination pipeline. Features are processed for each modality separately (Section 3.4, Section 3.5 and Section 3.6) and later merged by using ensembles of predictive clustering trees (Section 3.7). References: [10,11,12].

**Figure 3 sensors-22-07962-f003:**
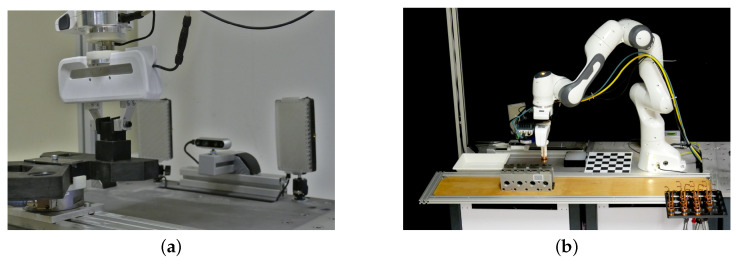
Experimental setup for testing exception strategy learning using multimodal data. (**a**) Setup for the *Cranfield benchmark*. (**b**) Setup for the copper ring insertion task.

**Figure 4 sensors-22-07962-f004:**
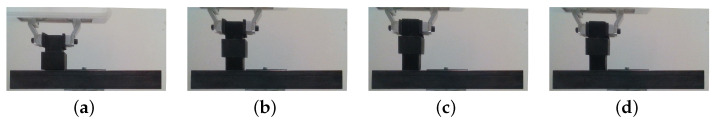
Different outcomes of the PiH task. (**a**) Successful insertion with correct parameters. (**b**) Insertion failure due to grasping error. (**c**) Insertion failure due to a positional error in the *x*-direction. (**d**) Insertion failure due to a positional error in the *y*-direction.

**Figure 5 sensors-22-07962-f005:**
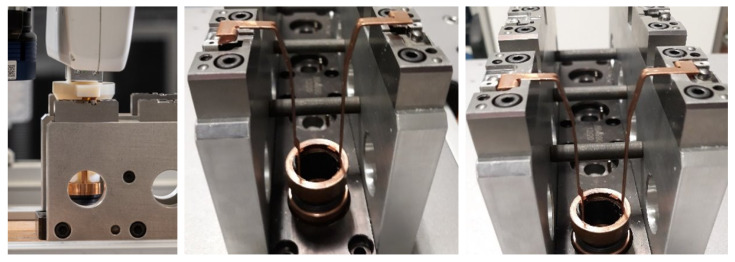
Left: Insertion of the ring into a modeling fixture. Center: Incorrect insertion. Right: Correct insertion.

**Figure 6 sensors-22-07962-f006:**
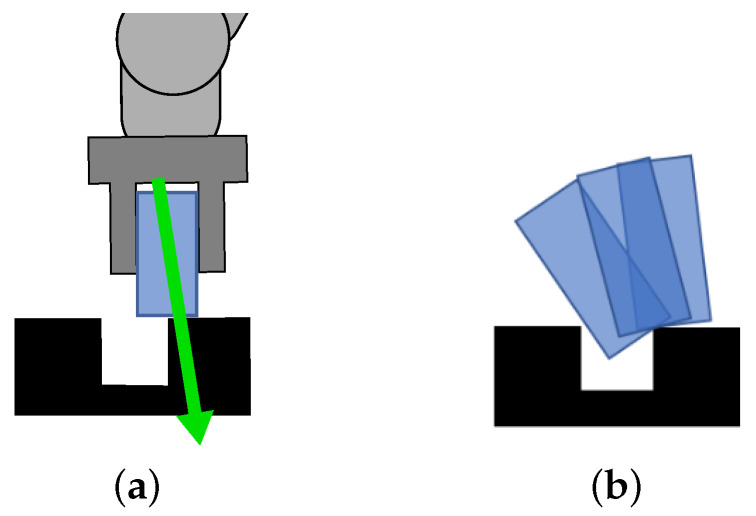
A scheme depicting contact point estimation (**a**) and another example where the grasped part comes into contact with the environment at the same point (**b**). The robot and the gripper are represented by the dark gray shape, whereas the grasped part and the environment are shown in blue and black, respectively. The contact vector estimate is shown with a green arrow.

**Figure 7 sensors-22-07962-f007:**
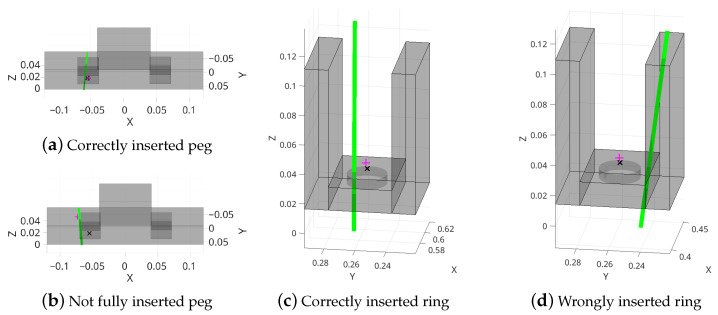
Contact vector estimation in four examples of the considered assembly tasks. A schematic model of the Cranfield base plate and the casting mold is shown in (**a**–**d**), respectively. The green line shows the contact vector, whereas the pink plus symbol shows the robot’s tool center point (TCP) at the time of contact and the black cross shows the target reference position.

**Figure 8 sensors-22-07962-f008:**
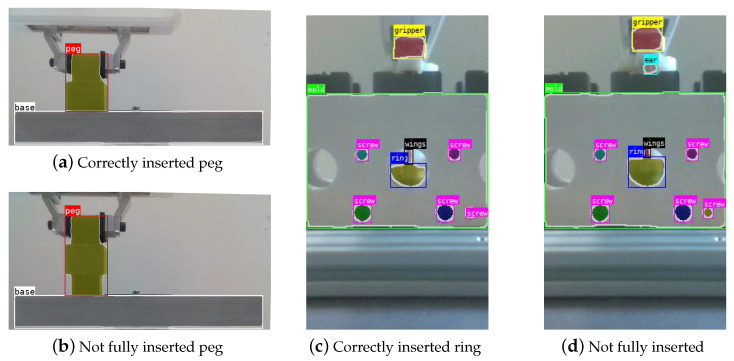
Bounding boxes and masks obtained by YOLACT using snapshots of the outcome of both tasks as input. For the PiH task (**a**,**b**), the base plate and peg are detected, regardless of the position/occlusion of the latter. In the copper ring insertion task (**c**,**d**), the gripper, mold, screws, ring base, wings, and ears are detected. Notice that ears are only detected when the part is not fully inserted.

**Figure 9 sensors-22-07962-f009:**
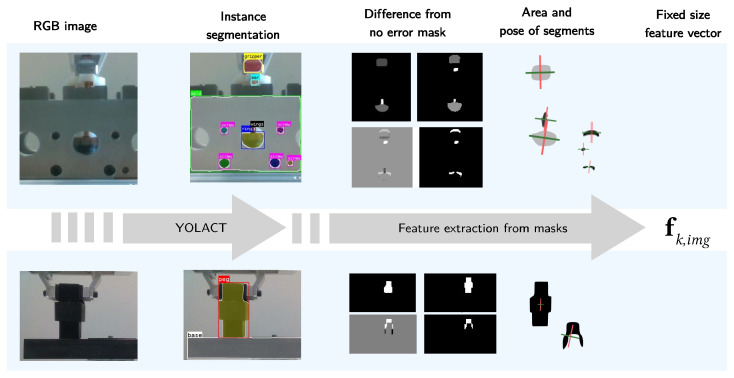
Image features extraction pipeline. The RGB snapshot of a situation is processed by a YOLACT model to obtain masks of different parts of interest. The obtained masks are then processed to obtain a low-dimensional fixed-size feature vector.

**Figure 10 sensors-22-07962-f010:**

The extracted peg mask for different executions of the copper ring insertion task. Red and green lines show the principal directions and determine the mask’s orientation. (**a**) $\Delta p_{x}$ = −10 mm; (**b**) $\Delta p_{x}$ = −5 mm; (**c**) $\Delta p_{x}$ = 0 mm; (**d**) $\Delta p_{x}$ = 5 mm; (**e**) $\Delta p_{x}$ = 10 mm.

**Figure 11 sensors-22-07962-f011:**
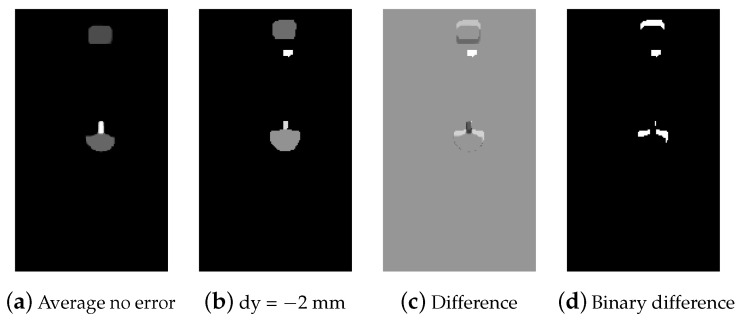
From left to right: (**a**) average segmentation mask for the successful copper ring insertion attempts, (**b**) segmentation mask for a failed insertion attempt (positional error in the *y*-direction), (**c**) difference of the segmentation masks, and (**d**) positive part of the difference.

**Figure 12 sensors-22-07962-f012:**
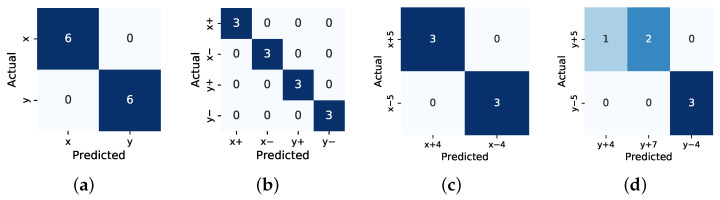
Confusion matrices for classification at different levels of the hierarchy. (**a**) First level of hierarchy (*x* or *y* displacement). (**b**) Second level (negative or positive displacement). (**c**) Third level (magnitude of *x* displacement); (**d**) Third level (magnitude of *y* displacement).

**Figure 13 sensors-22-07962-f013:**
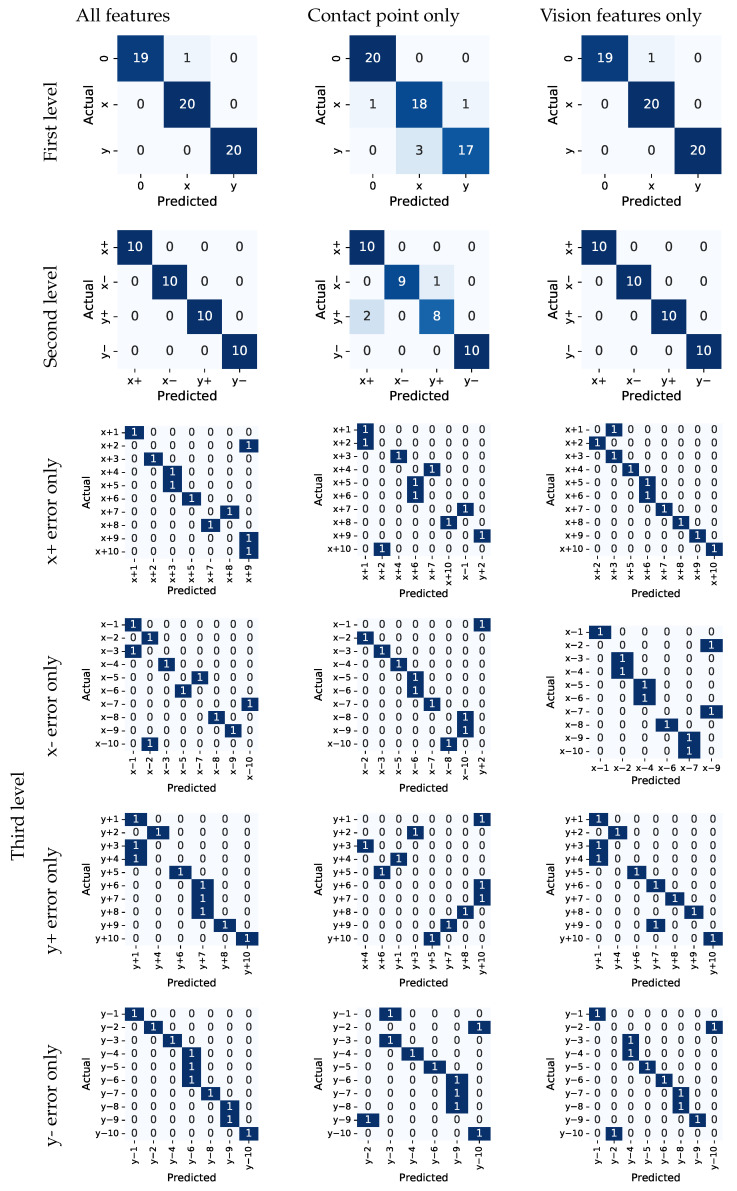
Confusion matrices for the PiH use case.

**Figure 14 sensors-22-07962-f014:**
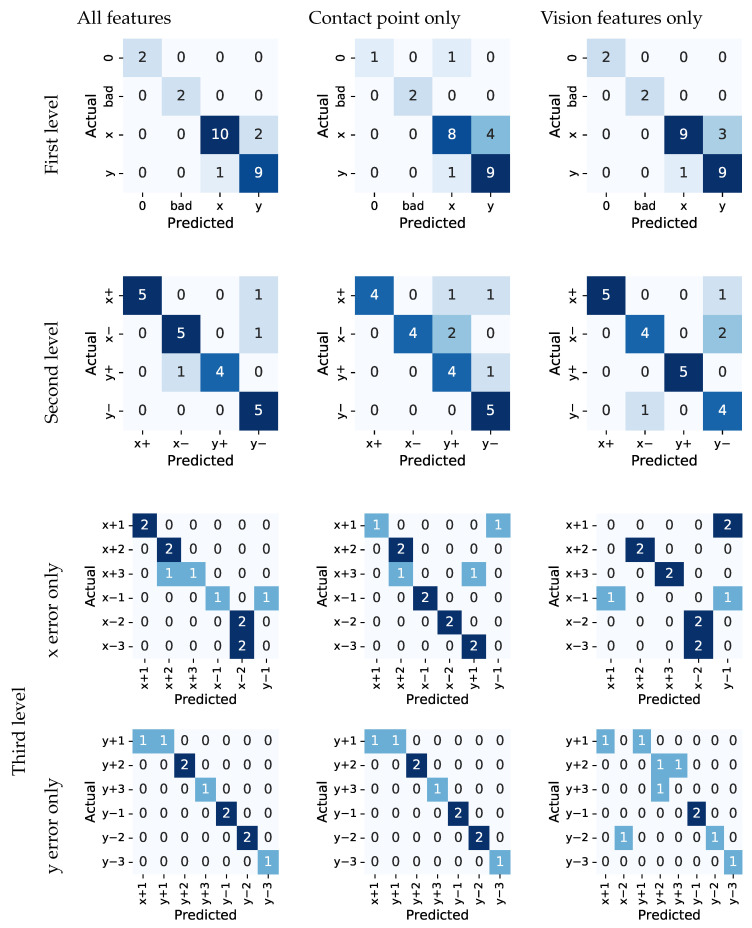
Confusion matrices for the copper ring insertion task.

## Data Availability

The data analyzed in this study are openly available in https://doi.org/10.5281/zenodo.7221443 (accessed on 13 October 2022) and https://doi.org/10.5281/zenodo.7221387 (accessed on 13 October 2022).

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
