# Peer review of "Determining Exception Context in Assembly Operations from Multimodal Data"

_sensors, 2022, doi:10.3390/s22207962_

Round 1

Reviewer 1 Report

The main innovation points of the paper are not clear, and there is a lack of experiments in real scenes and comparisons with related works. The content of the paper is written according to the specific scene, the method and the experiment are somewhat indistinguishable, and the generality of the method needs to be considered.

Author Response

We thank the reviewer for evaluating the work. We address the raised concerns point-by-point.

Point 1: The main innovation points of the paper are not clear, and there is a lack of experiments in real scenes and comparisons with related works.

Response 1: Automatic error recovery is an important issue in automation that has not yet been satisfactorily addressed. The exception strategy framework is a novel way to address this issue by exploiting human-robot collaboration aspects.

We revised the abstract to emphasize the framework's central pillar, context determination. The approach toward context determination is completely novel. To our best knowledge, no other work addresses the problem of context determination in robotics from multi-modal data in a similar manner.

Besides evaluation with the peg-in-hole benchmark, we evaluated the approach in a realistic setup with parts from a real industrial use-case, as shown in Figure 3(b). Additionally, we briefly evaluated the entire system in these scenarios. However, the paper's focus remains on methods for context determination and their detailed evaluation.

The above-discussed issues are addressed in the revised manuscript.

Point 2: The content of the paper is written according to the specific scene, the method and the experiment are somewhat indistinguishable, and the generality of the method needs to be considered.

Response 2: The answer is in two parts. The first part explains how the overall framework can be applied to a variety of scenarios, while the second focuses on the generalization of the method for context determination, which is the paper's main focus.

The overall exception strategy learning framework is general, but the concrete solution is, of course, case specific. Our scheme is to record a new exception strategy (recovery & resolve action) only when an error occurs and put it in the database associated with the context estimate. The main difference between related approaches is that we do not need to anticipate all possible errors in advance but update the knowledge base when necessary and it is closely related to the concept of "incremental learning".

Although we have set up a specific scene to evaluate the proposed method of context determination, our framework can be applied to a huge variety of assembly scenarios and beyond.

The method for context determination is evaluated on two different use cases. The first use-case, peg-in-hole task, is chosen specifically as it is generally considered to encompass a typical level of complexity for an industrial assembly task. Furthermore, we show that it is possible to apply the method to other tasks by performing evaluation in a real industrial use-case setup.

In order to apply the approach to a different use-case no changes in the procedure itself are needed thanks to the hierarchical structure of the approach. Apart from the identification of new error classes (can be something else than just position errors and bad/damaged part), only the instance segmentation model needs to be retrained.

Additional text was added in the introduction to highlight this aspect properly.

Point 3: English language and style are fine/minor spell check required

Response 3: We carefully checked the paper and corrected typos and grammar mistakes.

Reviewer 2 Report

The authors introduced '' Determining Exception Context in Assembly Operations from Multimodal Data”. The manuscript is written in clear and understandable language. It's a very interesting problem and an interesting presentation. The research certainly contributes to the subject area. In conclusion, the presented work has been prepared very well. I can recommend this article for publication after minor revisions. These are the following suggestions/recommendations for the authors:

·  Is this system new? Or it has been developed based on another original system, please show that in the abstract section.

·  What numerical techniques were used in the whole article investigation; this can be inset in the article?

·  For robots like in this manuscript, the main issue is the accuracy and stability of the robot for doing the tasks, so this issue must be inserted in the manuscript.

·  The time efficiency of the robot responsibility should be taken into your account.

·  External disturbances can be considered for checking the performance efficiency of the proposed algorithm.

·  Some of the equations are not derived, so the authors have to quote the source of the expression.

·  The authors should compare their work with recent similar articles in the subject area.

·  The developed algorithm(s) should be presented by summary flowcharts only.

·  The authors should be cited more recent references.

Author Response

Point 1: The authors introduced '' Determining Exception Context in Assembly Operations from Multimodal Data”. The manuscript is written in clear and understandable language. It's a very interesting problem and an interesting presentation. The research certainly contributes to the subject area. In conclusion, the presented work has been prepared very well. I can recommend this article for publication after minor revisions.

Response 1: We thank the reviewer for positive evaluation of the work. We followed the reviewer’s suggestions and adapted the article accordingly.

Point 2: Is this system new? Or it has been developed based on another original system, please show that in the abstract section.

Response 2: The overall exception strategy learning framework was previously introduced. However, this paper deals specifically with one component of the framework, i.e. context determination, as clearly stated in the abstract. The approach to context determination is completely novel and contributes towards better performance of the overall exception strategy framework. We adapted abstract and the introduction to highlight this aspect better.

Point 3: What numerical techniques were used in the whole article investigation; this can be inset in the article?

Response 3: We added a brief summary of the methods used at the beginning of Section 3 to address this point.

Point 4: For robots like in this manuscript, the main issue is the accuracy and stability of the robot for doing the tasks, so this issue must be inserted in the manuscript.

Response 4: To perform the assigned tasks, we applied a passivity-based impedance controller for manipulators with flexible joints. We assume that the controller parameters were carefully tuned to ensure stable and compliant operation in unstructured environments, where we can expect deviations in task parameters. This is also noted in Section 3.1 of the revised manuscript.

Point 5: The time efficiency of the robot responsibility should be taken into your account.

Response 5: Using the presented framework does not prolong the execution time of successful task executions as the time to obtain context information is negligible. No computationally expensive operations are involved in the processing of the force-torque data. We employ a state-of-the-art YOLACT system for the image segmentation step, praised for its high interference step.

For unsuccessful attempts, based on the determined context, the framework tries to generate new policies that first resolve the issue and then repeat the task with refined policy parameters. Our brief experimental evaluation in the robotic work cell has shown that in most cases, the problem can be resolved in one iteration. We addressed this aspect in Section 4.3 of the revised manuscript.

Note that the quality of the demonstrated policies was not the primary focus of this paper. However, in our previous research, we developed methods suitable for the optimization of the velocity profile of the demonstrated policies with RL and ILC.

See: B. Nemec, A. Gams, and A. Ude (2013) Velocity Adaptation for Self-Improvement of Skills Learned from User Demonstrations, 13th IEEE-RAS International Conference on Humanoid Robots (Humanoids), Atlanta, Georgia, pp. 423-428.

Point 6: External disturbances can be considered for checking the performance efficiency of the proposed algorithm.

Response 6: We agree that evaluating the exception strategy learning framework against external disturbances would be an excellent performance criterion when assessing the entire system. However, in this article, we focused only on developing methods for determining the context from multi-modal data. Therefore, we believe that an in-depth evaluation of the entire framework is beyond the scope of this paper.

Point 7: Some of the equations are not derived, so the authors have to quote the source of the expression.

Response 7: We assume that the equations of concern were (2) and (6). We added additional explanations to clarify how they are linked to other steps.

We also omitted Eq. (13) in the revised version of the script, as the expression is not directly used in the remainder of the text and does not contribute to understanding the text.

Point 8: The authors should compare their work with recent similar articles in the subject area.

Response 8: To our best knowledge, this is the first approach that employs ensembles of predictive clustering trees to address the problem of context determination from multimodal data in robotics. In Section 2, we mention that “combining visual and contact information has been investigated in multiple works in robotics”, but not in the context of automatic error recovery. We share the idea of combining data from visual and force-based sensing, but want to develop these concepts “further towards a structured representation of the task context in order to develop an integrated solution for the automatic handling of failures in assembly processes.” We slightly adapted Section 2 to highlight this difference.

Point 9: The developed algorithm(s) should be presented by summary flowcharts only.

Response 9: A simplified pipeline for context determination is presented in Figure 2. We updated the caption to advise the reader where to find further details about the respective steps.

Point 10: The authors should be cited more recent references.

Response 10: Approximately half of the citations are from the period of last five years.

Assembly is intensively investigated research area from the early stage of robotics. Therefore, we deliberately kept some older citations from this field to emphasize the relevance of the related investigations.

We additionally cited a recent review article on PiH assembly: J. Jiang et.al. (2020). State-of-the-Art control strategies for robotic PiH assembly. Robotics Comput. Integr. Manuf. 65 : 101894.

We also omitted some older references about the application of ensemble learning methods in other domains, and replaced them with a more recent and highly cited survey article: X.Dong, et. al. (2020) A survey on ensemble learning. Front. Comput. Sci., 14(2), pp. 241‒258.

Reviewer 3 Report

The paper proposes an approach to generate unified low-dimensional context descriptions based on multimodal-data, which can be used for learning of exception strategies in various assembly tasks. Experiments show the effectiveness of the approach.

Author Response

We thank the reviewer for positive evaluation of our work.

Reviewer 4 Report

The paper is well-written and the proposed approach adequately described.

My only concern is about the choice of the RGB-D camera Intel RealSense. I suggest to specify the model (from Figure 3a it looks D415 or D435 to me) and to better explain of which depth acquisition technology is equipped. Depth acquisition technologies is core in RGB-D cameras and should be carefully chosen according to the final application, as described for instance in Moos et al., "Analysis of RGB-D camera technologies for supporting different facial usage scenarios."

Author Response

We thank the reviewer for positive evaluation of our work.

Point 1: My only concern is about the choice of the RGB-D camera Intel RealSense. I suggest to specify the model (from Figure 3a it looks D415 or D435 to me) and to better explain of which depth acquisition technology is equipped.

Response 1: We used a D435i camera in the experiments and specified this in the revised manuscript. Please note that we did not use depth data at this research stage. However, as pointed out in the discussion, we plan to use depth data in future work.

Round 2

Reviewer 1 Report

The author answered my queries and the paper can be accepted.